# Analysis of the Calculation Method for the Thermal Transmittance of Double Windows Considering the Thermal Properties of the Air Cavity

**Minjung Bae [1,2]**, **Youngjun Lee [3]**, **Gyeongseok Choi [1]**, **Sunsook Kim [2]** and **Jaesik Kang [1,\*]**

[1]   Department of Living and Built Environment Research, Korea Institute of Civil Engineering and Building Technology, Goyang 10223, Korea; baeminjung@kict.re.kr (M.B.); bear717@kict.re.kr (G.C.)
[2]   Department of Architecture, College of Engineering, Ajou University, Suwon 16499, Korea; kss@ajou.ac.kr
[3]   Institute of Environmental Building Facade Engineering, Bel Technology Co. Ltd., Seoul 05548, Korea; leeyj@beltec.co.kr
\*   Correspondence: jskang@kict.re.kr; Tel.: +82-31-910-0353

**Abstract:** The calculation method for the thermal transmittance (U-value) of double windows as specified by the Korean government (ISO 15099) is often inappropriate. To develop a more suitable calculation method, the thermal properties of the air cavity between the internal and external windows should be considered. Herein, seven cases of double windows were set up. The air cavities were designed in accordance with international standards and computational fluid dynamics (CFD) and used for the calculation of the U-values of the double windows according to ISO 15099 and 10077. All the calculated U-values were compared with experimentally obtained values. In accordance with the ISO 10077-1 method, the thermal resistance of the air cavity calculated using CFD could produce double window U-values that are similar to the experimentally obtained values. In most cases, the difference between the theoretical and experimental U-values was 5% and less than 0.14 W· m$^{-2}$· K$^{-1}$, implying that the U-values calculated using CFD and the ISO 10077-1 method are approximately equal to the experimentally obtained U-values. Korean regulations do not include ISO 10077-1 for double-window assessment. However, these criteria can provide a solution in improving the accuracy of the calculation of the overall thermal transmittance of double windows.

**Keywords:** energy labeling program for windows; double windows; overall thermal transmittance of windows

## 1. Introduction

Since 2012, the Korean government has operated the Energy Efficiency Standards and Labeling Program for windows, which requires window companies to provide the energy ratings for their products prior to sale. The grades, on a scale of 1–5, are determined based on the test results of the thermal transmittance (U-value), airtightness, and thermal resistance of the windows and doors [1,2] according to Korean Standards KS F 2278 and KS F 2292, respectively. The government has suggested a simulation method in the program for the determination of the thermal performance of windows. This method provides an alternative procedure by which window companies can save time and money on laboratory testing, which is necessary for the determination of energy ratings. Following this method, window companies prepare a window product with a determined energy rating and conduct the simulation evaluation to review the validity of the base model. If the difference between the experimental and theoretical values obtained using the base model does not exceed a range specified in the operational regulations [3], the base model can be implemented to develop a series model. The series model is a partial modification of the base model, which generally changes the glazing system or the

thermal break in the window frame. This means that the thickness of the glazing system on the base and series models should be the same. Using the regulations, window companies can get the certified thermal transmittance required of their products faster and at a cheaper cost. The Korean government allows window companies to use the calculation method proposed by the International Organization for Standardization (ISO), standard 15099 [4]. Therefore, WINDOW/THERM [5] is commonly used as a simulation program to evaluate the thermal performance of windows. In a previous study [6], we analyzed the origin of the differences in the calculation results of the thermal performance of a window depending on a simulator and suggested a possible solution. However, window companies are reluctant to use the calculation method because the results obtained using the method are different from those obtained experimentally. The thermal performance of a single window can be calculated according to ISO 15099 such that it does not vary much from the test value [7]. However, this method cannot be used for a double window because of the thickness of the air cavity in the direction of heat flow. Double windows are a common window type in Korea [8], and they are mainly used in residential buildings. These windows consist of four windows that open horizontally in one window frame and have an air cavity between the external and internal windows. The thickness of the air cavity is usually 70−120 mm, which means the length of the heat flow direction. If this thickness exceeds 50 mm, ISO 15099 requires that another calculation method should be used to determine the thermal properties of the air cavity, for example, performing laboratory tests. In a previous study [9], to validate the ISO 15099 method, the thermal properties of the air cavity between internal and external windows were calculated based on computational fluid dynamics (CFD) and ISO 15099. It was observed that the ISO 15099 method was inappropriate for calculating the thermal properties of the air cavity under actual experimental conditions. Therefore, it is necessary to use another method to determine the thermal characteristics of the air cavity between the internal and external windows in a double window to indicate the circumstances of an experimental test. Furthermore, when determining the thermal properties of the air cavity using the ISO 15099 method, it is assumed that the double window is part of a glazing system. This method assumes a double window to be a single window with a huge thick glazing system and a window frame. For these reasons, window companies suspect the reliability of the ISO 15099 method and require a more suitable method for calculating the U-values of double windows.

In this study, to determine an appropriate calculation method, ISO 15099 and ISO 10077 were used in the calculation of the thermal transmittance of double windows. Given that it is relevant to select a calculation method that is appropriate for determining the thermal properties of the air cavity between the internal and external windows in a double window, first, the U-values of double windows were calculated using WINDOW/THERM, based on Korean regulations. Thereafter, series ISO 10077-1 [10] and 10077-2 [11] of ISO 10077 were also employed to calculate the U-values of the double windows. Specifically, ISO 10077-1 specifies a method for the calculation of the thermal transmittance of a double window, whereas 10077-2 provides reference input data for the calculation of the thermal transmittance of frame profiles as well as the linear thermal transmittance of their junction with glazing. Seven cases of double windows, including four types of double window products and six types of glazing systems, were considered. The thermal properties of the air cavity in each case were determined using International Standards and were simulated using CFD. Finally, the U-values computed using ISO 15099 and ISO 10077-1 were compared with the experimental results.

## 2. Methods

### 2.1. Double Window Types

Table 1 lists the six types of glazing systems that are available for double windows. The glazing systems were selected based on the International Glazing Database (IGDB), which is operated by the National Fenestration Rating Council (NFRC). These consist of two 5 mm glass panes separated by a 12 mm-wide gap filled with air (Air) or argon gas (Ar). In Table 1, LE and CL correspond to glass

panes with and without low-emissivity coating, respectively. The numbers before each abbreviation correspond to the thickness of the glass pane or that of the gap between glass panes.

**Table 1.** Glazing systems for double windows.

| Glazing System | Composition | Ug (W·m$^{-2}$·K$^{-1}$) |
|---|---|---|
| A | 5CL + 12Air + 5CL | 2.901 |
| B | 5CL + 12Air + 5LE | 1.704 |
| C | 5CL + 0.76PVB + 3CL + 12Air + 5LE | 1.664 |
| D | 5CL + 12Air + 5LE | 1.624 |
| E | 5LE + 12Ar + 5LE | 1.278 |
| F | 6LE + 14Ar + 5CL | 1.124 |

The glazing systems B and D have glass pane coatings of thickness 5 mm, with emissivities of 0.035 and 0.026, respectively. It causes that thermal the performances of the glazing system B and D are different. The glazing system C is 25.76 mm thick because it comprises one laminated glass pane. It consists of 5 and 3 mm thick clear glass panes and a 0.76 mm thick polyvinyl butyral (PVB) coating in-between. Glazing system F is 25 mm thick and consists of a 14 mm-wide gap filled with argon gas between one 6 mm-coating glass pane and one 5 mm glass pane.

Table 2 indicates the seven double window cases according to the product name and the glazing system. In this study, three double window products with polyvinyl chloride (PVC) frame and one double window product with aluminum frame were chosen. These window products, which are the horizontal slide type, are widely available in the Korean market. That with product name VBF250 has an external window that consists of an upper component that slides and a lower component that is fixed. The others have internal and external windows that slide, which are common in Korea. Each type of double window has a different frame profile, thus, they can have different distances between the external and internal windows. Products S3-235 and S5-250 have the same distance (88 mm) between the external and internal windows, while for HS235D, the distance is 94 mm. Product VBF250 has a different upper and lower component in the external window, so the distance between its external and internal windows is 70.6 mm in the upper part and 94.5 mm in the lower part. Cases 1 and 2 are the same type of double window with two different glazing systems and so are Cases 3 and 4. Cases 6 and 7 are based on VBF250, and the upper part of the external window and the internal window have glazing systems A or B in each case, but the lower part of the external window is the same.

**Table 2.** Specification on the double window cases.

| Case | Product Name | Frame Material | Glazing System | | Distance between the External and Internal Windows (mm) |
|---|---|---|---|---|---|
| | | | External Window | Internal Window | |
| 1 | <S3-235> | PVC | A | D | 88 |
| 2 | | | E | E | 88 |
| 3 | <S5-250> | PVC | A | D | 88 |
| 4 | | | A | A | 88 |

**Table 2.** *Cont.*

| Case | Product Name | Frame Material | Glazing System | | Distance between the External and Internal Windows (mm) |
|------|-------------|---------------|----------------|----------------|----------------|
| | | | **External Window** | **Internal Window** | |
| 5 | <HS235D> | Aluminum | F | F | 94 |
| 6 | <VBF250> | PVC | Upper　B<br>Lower　C | B | 70.6<br>94.5 |
| 7 | | | Upper　A<br>Lower　C | A | 70.6<br>94.5 |

## 2.2. Laboratory Tests

Laboratory tests were carried out on all the double window products according to KS F 2278. The test equipment consisted of a 2.0 × 2.0 m attachment frame to which the test specimen was attached and cold and hot chambers, which each included a cold wind blower and a heater box, respectively. The attachment frame was fixed between the cold and hot chambers, and the air temperature of the cold chamber was set to 0 °C, while that in the hot chamber and the heater box was set to 20 °C. The equipment was operated until the two chambers reached a steady state after which the temperature and quantity of heat in each chamber and the heater box were measured three times every 30 min. During this test, the steady state implied that the air temperature and surface temperature were kept constant, and the variation in the difference in the air temperature between the heater box and the cold chamber was within 3% per hour. To measure the surface temperature of the hot and cold sides, each specimen was divided into nine areas, and a T-type thermocouple was attached to the center of each of the nine areas [9], represented by the orange dots in Figure 1. In this study, eight additional T-type thermocouples, represented by the green dots in Figure 1, were installed in the corner area of the glazing systems, 3 cm away from the window frame. Figure 1a shows the locations at which the thirteen T-type thermocouples were installed on the one side of a specimen, these were applied to the surfaces of the internal and external windows, as shown in Figure 1b.

## 2.3. Calculation of the Thermal Resistance of the Air Cavity between the Windows

The thermal resistance of the air cavity between the internal and external windows impacts the calculation of the thermal transmittance of double windows, as the computed U-value should be similar to the experimental result. Table 3 denotes the thermal resistance of the air cavity in each window product. The thermal resistance of the air cavity in each case can be calculated using three methods, i.e., ISO 15099, ISO 10077-1, and CFD. ISO 15099 allows for the calculation of the effective conductivity of the unventilated frame cavity, and is defined according to the thickness or width of the air cavity in the direction of heat flow. In this study, WINDOW/THERM was chosen for computing the thermal properties of the air cavities in the double windows according to ISO 15099. Thus, the thermal resistance of the air cavity was calculated using the effective conductivity and thickness. In this software, the air cavity between the external and internal windows was considered as a wide gap between the glass panes, i.e., the air cavity presumably belongs to a 132 mm thick giant glazing system, consisting of four pane glass and three gaps filled with air or argon.

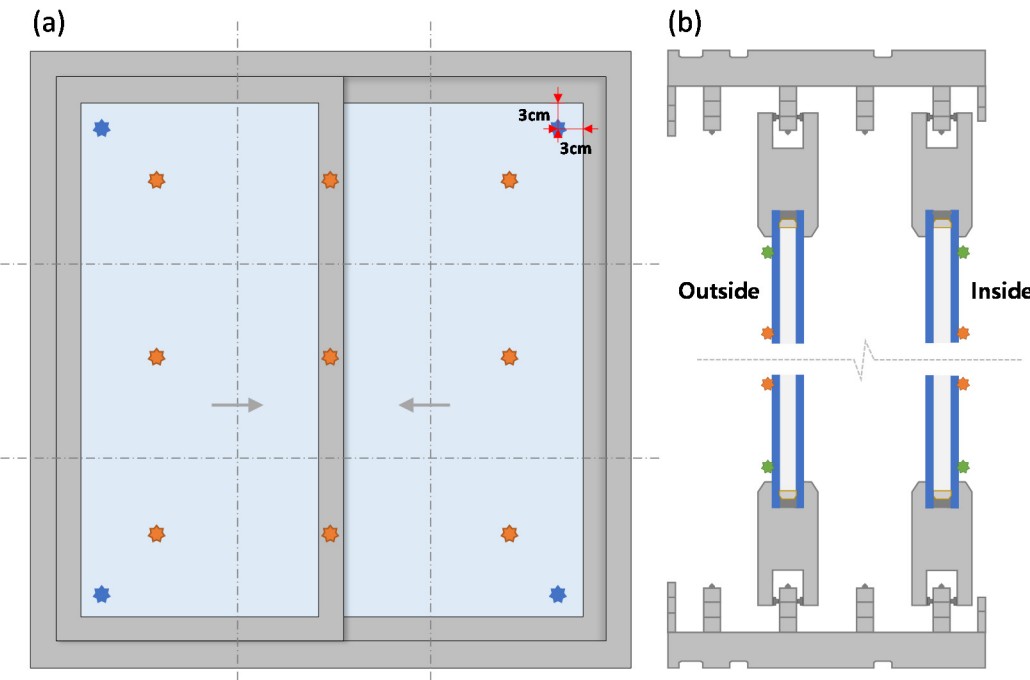

**Figure 1.** (**a**) Locations of T-type thermocouples on one side of a specimen, and (**b**) sections of specimens with T-type thermocouples on the internal and external windows.

However, in ISO 10077, the calculation of the overall thermal transmittance of the double windows is based on component parts, the elements constituting the glazing systems, the thermal transmittance of the frame, and the linear thermal transmittance of the frame/glazing junction. Specifically, ISO 10077-1 provides the values of the thermal resistance of unventilated air cavities in double windows according to the thickness of the air gap (6, 9, 12, 15, and 50 mm) in the form of a table. These values depend on whether the glazing status on one side has a normal emissivity coating or is uncoated. Unfortunately, there is no exact value for the thermal resistance of air cavities with thicknesses in the range 70.6–94 mm in ISO 10077-1. As previously reported [9], CFD was used to analyze the actual thermal characteristics of the air cavities. The CFD model is not a precise simulation of the air cavity between the external and internal windows. However, it aims to analyze the thermal properties stemming from the actual width and height. Based on CFD, Cases 1–5 imitating the air cavity in a double window were modeled as a closed 1 × 2 m air cavity (width and height, respectively). The thicknesses, measured in the horizontal direction of the heat flow based on the distance between the external and internal windows, are specified in Table 3. Two 1 × 2 m air cavities (width and height, respectively), were formed in a double window because the window was divided. For this reason, the two air cavities exhibited symmetrical air flows and temperature distributions. In this study, it was assumed that the CFD results obtained for one air cavity can be applied to all the air cavities in a double window. The ambient temperature of the CFD air cavity model were 0 and 20 °C, and these were defined by the external and internal glazing systems for the surface emissivity and surface heat transfer coefficient of the CFD model. Cases 6 and 7 had an exterior window divided into four sides. Therefore, the CFD model for these cases had two types of exterior boundary conditions and two different thicknesses. The upper and lower parts of the air cavity had different boundary conditions owing to the different glazing systems in the double windows. However, the air cavities exhibited symmetrical thermal properties. All the values of the thermal resistance computed using CFD were higher than those calculated using ISO 15099. This explains why the laboratory test values of the thermal transmittance of the double windows are lower than those obtained theoretically [9].

**Table 3.** Thermal characteristics of air cavities for the double window cases.

| Case | Glazing System | | | Distance between the External and Internal Windows (mm) | Effective Thermal Conductivity ($W \cdot m^{-1} \cdot K^{-1}$) | Thermal Resistance ($m^2 \cdot K \cdot W^{-1}$) | | |
|------|------|------|------|------|------|------|------|------|
| | External Window | | Internal Window | | | ISO 15099 ($R_i$) | ISO 10077-1 ($R_s$) | CFD |
| 1 | A | | D | 88 | 0.4333 | 0.203 | | 0.219 |
| 2 | E | | E | 88 | 0.4252 | 0.207 | | 0.219 |
| 3 | A | | D | 88 | 0.4333 | 0.203 | | 0.219 |
| 4 | A | | A | 88 | 0.4577 | 0.194 | | 0.219 |
| 5 | F | | F | 94 | 0.4489 | 0.209 | N/A | 0.220 |
| 6 | Upper | B | B | 70.6 | 0.3475 | 0.203 | | 0.216 |
| | Lower | C | | 94.5 | 0.4649 | 0.203 | | |
| 7 | Upper | A | A | 70.6 | 0.3607 | 0.196 | | 0.216 |
| | Lower | C | | 94.5 | 0.4867 | 0.194 | | |

### 2.4. Calculation of the Thermal Transmittance of the Double Windows

Generally, the overall thermal transmittance of a window can be calculated using WINDOW/THERM according to ISO 15099 and using the thermal resistance values of the air cavity obtained via CFD based on ISO 10077. Unfortunately, the thermal properties of an air cavity according to ISO 10077-1 are not available for all the seven cases used in this study. The ISO 10077-1 methodology includes the calculation of the linear thermal transmittance of the double window. The corresponding values could be obtained directly from a table in ISO 10077-1 or calculated from a formula in ISO 10077-2.

ISO 15099 includes the procedure for calculating thermal transmittance. In this procedure, the effect of three-dimensional heat transfer in frames and glazing units is not considered. Additionally, in this procedure, the linear thermal transmittance and frame thermal transmittance, $U_f$, was calculated. However, there is an alternative procedure that can be used to calculate these values, which is used in area-based calculations and by WINDOW/THERM. In this case, Equation (1) was used to calculate the total thermal transmittance:

$$U_t = \frac{\sum U_{cg}A_c + \sum U_{fr}A_f + \sum U_{eg}A_e + \sum U_{div}A_{div} + \sum U_{de}A_{de}}{A_t} \tag{1}$$

With this method, it is unnecessary to determine the linear thermal transmittance. Instead, the glass area, $A_{gv}$, is divided into the center-glass area, $A_c$, plus the edge-glass area, $A_e$. Similarly, the thermal transmittance of the glazing system is divided into the center-glass and edge-glass systems, $U_{cg}$ and $U_{eg}$, respectively, which are used to characterize each glass system area. If dividers are present, then the divider area, $A_{div}$, and the divider thermal transmittance, $U_{div}$, were calculated along with the corresponding divider edge area, $A_{de}$, and thermal transmittance, $U_{de}$. $U_{eg}$ can be determined from the following equation:

$$U_{eg} = \frac{\Phi_{eg}}{l_{eg}(T_{ni} - T_{ne})} \tag{2}$$

where $l_{eg}$ is the length of the edge of the glass area and is equal to 63.5 mm. These lengths were measured from the internal side. The quantity, $\Phi_{eg}$, represents the heat flow rates through edge-glass areas (internal surfaces), including the effect of glass and spacer, and it is expressed in units, per length of edge-glass. In WINDOW/THERM, the thermal resistance of an air cavity between internal and external windows was used to calculate the thermal transmittance of the glazing system, given that the air cavity was assumed to be a component of the glazing system. The thermal transmittance of the glazing system, according to ISO 15099, can be determined from the following equation:

$$U_g = U_{cg} + U_{eg} = \frac{1}{R_t} \tag{3}$$

where $R_t$ is obtained by adding the thermal resistances at the external and internal boundaries and of the glazing cavities and layers.

$$R_t = \frac{1}{h_{ex}} + \sum_{i=2}^{n} R_i + \sum_{i=1}^{n} R_{gv,i} + \frac{1}{h_{int}}. \tag{4}$$

Figure 2 shows the numbering scheme of the glazing system. Specifically, the thermal resistance of the $i$th glazing is given by:

$$R_{gv,i} = \frac{t_{gv,i}}{\lambda_{gv,i}} \tag{5}$$

and the thermal resistance of the $i$th space is given by:

$$R_i = \frac{T_{f,i} - T_{b,i-1}}{q_i} \tag{6}$$

where $T_{f,it}$, and $T_{b,i-1}$ are the external and internal facing surface temperatures of the $i$th glazing layer, respectively. It should be noted that the first space corresponds to the external environment, the last space corresponds to the internal environment, and the spaces in between correspond to the glazing cavities. Therefore, Equation (6) gives the thermal resistance of an air cavity between external and internal windows in a double window using ISO 15099.

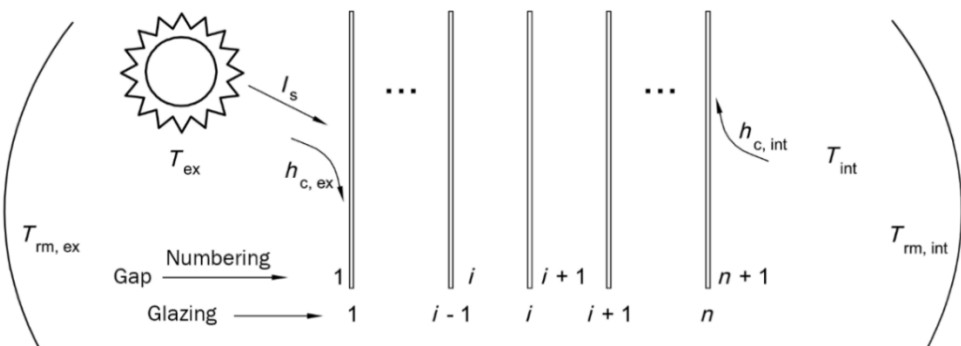

**Figure 2.** Numbering scheme for glazing system layers.

However, the U-value of a double window according to the ISO 10077 method needs linear thermal transmittance. The preferred method of establishing the values of the linear thermal transmittance was by numerical calculations using the formulas included in ISO 10077-2, Annex F. However, when the results of a detailed calculation are unavailable, ISO 10077-1 provides default values of the linear thermal transmittance for typical combinations of frames, glazing, and spacers.

ISO 10077-1 details the procedure for calculating the thermal transmittance, $U_w$, of a double window, which is a system consisting of two separate windows, as shown in Figure 3. For this method, it is necessary to calculate the thermal transmittances of the internal and external windows, $U_{w1}$ and $U_{w2}$, respectively. The thermal transmittance of a single window, $U_{w1}$ or $U_{w2}$, was calculated using the following formula:

$$U_w = \frac{\sum A_g U_g + \sum A_f U_f + \sum l_g \Psi_g + \sum l_{gb} \Psi_{gb}}{A_f + A_g} \tag{7}$$

where $U_g$ and $U_f$ are the thermal transmittances of the glazing system and frame, respectively, $\Psi_g$ is the linear thermal transmittance due to the combined thermal effects of glazing, spacer and frame, and $\Psi_{gb}$ is the linear thermal transmittance due to the combined thermal effects of glazing and glazing bar. The internal surface resistance, $R_{si}$, of the external window when used alone. The external surface resistance, $R_{se}$, of the internal window when used alone, and the thermal resistance, $R_s$, of the space between the glazing in the two windows were also calculated according to the given equations. In this study, the thermal resistance of each air cavity could not be defined by ISO 10077-1, whereas it could be defined by ISO 15099 and CFD, as shown in Table 3. The U-value of a double window could be calculated based on ISO 10077-1 using the thermal resistance of the air cavity computed using the CFD method. Then, the thermal transmittance of the double window was calculated using the following formula:

$$U_w = \frac{1}{U^{-w1} - R_{si} + R_s - R_{se} + U^{-w2}} \tag{8}$$

In this study, three methods were considered for calculating the thermal transmittance of double glazing, as shown in Table 4. Method A involves calculating the thermal resistance of an air cavity between external and internal windows, according to ISO 15099, and finally calculating the U-value of a double window in WINDOW/THERM software. In this method, the linear thermal transmittance does not need to be calculated. Methods B and C are based on the ISO 10077-1 methodology for determining the U-value of a double window. The thermal resistance of the air cavity computed by CFD, as shown in Table 3, is used as input data for these methods; however, Method B uses the default value of

the linear thermal transmittance in ISO 10077-1. Method C uses the linear thermal transmittance determined by numerical calculations using the formulas included in ISO 10077-2, Annex F. The three methods will be evaluated for the validity of whether the U-value of a double window similar to experimental values can be derived.

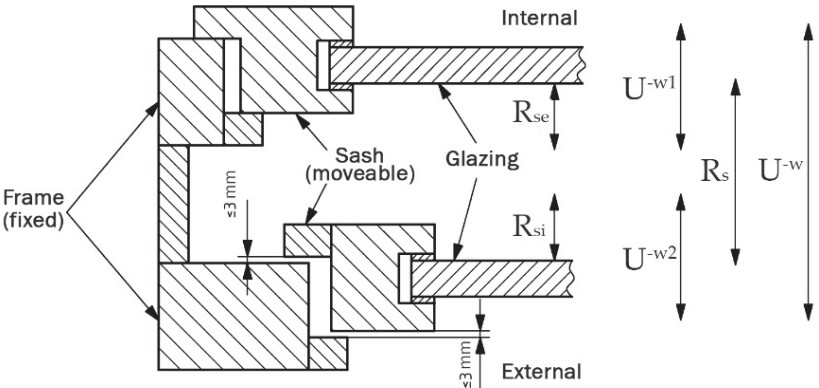

**Figure 3.** Illustration of a double window defined according to ISO 10077-1.

**Table 4.** Methods for calculating the U-value of a double window using three different sets of data.

| | Calculation Method for the Data | Method A | Method B | Method C |
|---|---|---|---|---|
| 1 | U-value of a double window | ISO 15099 | ISO 10077-1 | ISO 10077-1 |
| 2 | Thermal resistance of air cavity | ISO 15099 | CFD | CFD |
| 3 | Linear thermal transmittance | N/A | ISO 10077-1 (table) | ISO 10077-2 (calculation) |

## 3. Results and Discussion

Table 5 summarizes the thermal transmittance of seven double windows according to the three calculation methods and the laboratory tests. All the calculated U-values were higher than those obtained based on the laboratory tests, except for Case 1. According to Korean regulations, for U-values to be valid, the difference between the experimental and theoretically obtained values should not exceed 0.14 $\text{W·m}^{-2}\text{·K}^{-1}$. Calculation method A based on ISO 15099 exhibited a difference between experimental and theoretical U-values of 10% to 24%, although the Korean government operates the regulation to ensure the use of this method. Four cases were considered valid in Korean regulations, and three cases were considered invalid. The difference between the theoretical and experimental values was lower when ISO 10077-1 (Methods B and C), where the thermal resistance of the air cavity was calculated via CFD was used than when ISO 15099 was used. The CFD value used in this method was calculated using a simple air cavity model that depended on the actual height and width of the cavity. Nevertheless, the ISO 10077-1 method with CFD implementation (Method B and C) was considered valid based on Korean regulations because the difference between the experimental and calculated U-values was less than 0.14. However, it needs to be revised when determining the calculation method for the U-value of a double window to ensure the effective operation of Korean regulations. It should include the method that can reflect the thermal characteristics of the air cavity between the internal and external windows under experimental conditions, and this method should be able to present numerical results.

**Table 5.** The thermal transmittance of double windows (U-value) in the laboratory and the calculation methods.

| Case | Laboratory Test (W·m$^{-2}$·K$^{-1}$) | Method A (W·m$^{-2}$·K$^{-1}$) | Method B (W·m$^{-2}$·K$^{-1}$) | Method C (W·m$^{-2}$·K$^{-1}$) |
|---|---|---|---|---|
| 1 | 1.220 | 1.216 (−0.3%) | 1.147 (−6.0%) | 1.134 (−7.0%) |
| 2 | 0.737 | 0.915 (24.2%) | 0.872 (18.3%) | 0.845 (14.7%) |
| 3 | 1.113 | 1.227 (10.2%) | 1.142 (2.6%) | 1.129 (1.4%) |
| 4 | 1.314 | 1.470 (11.9%) | 1.355 (3.1%) | 1.335 (1.6%) |
| 5 | 1.006 | 1.152 (14.5%) | 1.030 (2.4%) | 0.991 (−1.5%) |
| 6 | 0.950 | 1.046 (10.1%) | 0.986 (3.8%) | 0.988 (4.0%) |
| 7 | 1.187 | 1.320 (11.2%) | 1.246 (5.0%) | 1.233 (3.9%) |

The method for calculating the linear thermal transmittance influenced the accuracy of the thermal transmittance of a double window. The U-value of the double window computed using the calculated linear thermal transmittance (Method C) was closer to the experimental value than that computed using the given linear thermal transmittance (Method B). U-values of Case 6 were similar, regardless of whether Method B or C was used, but the results obtained by Method C were closer to the experimental value than those obtained by Method B. This is because the default values of linear thermal transmittance, provided by ISO 10077-1, are considered conservatively. These values are larger than the actual linear thermal transmittance value, according to ISO 10077-2.

Although the same frame is used in Cases 1 and 2, Case 2 had a lower U-value than Case 1 because of the good thermal performance of the applied glazing system. The calculated U-values for Case 2 were most deviated from the experimental value compared to those for the other cases. It is thought that there are factors to be considered when calculating the U-value of a double window with excellent thermal performance of the glazing system. Therefore, in future studies, it would be necessary to identify the reasons behind this analysis, as well as the methods that should be included in the calculation.

## 4. Conclusions

The Korean government has been operating a simulation system for assessing the thermal performance of windows to allow companies to save time and money in determining window energy ratings. However, the uncertainty in the calculation results with respect to double windows has been discussed steadily. Thus, window companies are reluctant to use this calculating method. It has led window companies attempting to realize experimental U-values, even though the process is costly and time-consuming. According to Korean regulations, the procedure provided in ISO 15099 is used in the calculation of the thermal transmittance of double windows. However, our findings indicate that this method, ISO 15099 resulted in only four out of seven calculated values satisfying the criteria imposed by the Korean regulations. Further, all four valid values differ significantly from the experimental values. In a previous study [9], the importance of adopting the appropriate thermal properties of the air cavity between internal and external windows during the calculation of the thermal performance of double windows was reported. Therefore, the ISO 15099 method is no longer suitable for determining the thermal properties of the air cavity between internal and external windows, which is used to calculate the U-value of double windows. This method should be improved such that it can adopt the

thermal resistance of the air cavity under experimental conditions, and the CFD method used in this study is one of several methods that can be used. With the CFD method, it is possible to provide a table that can be used in calculating the U-values of double windows by pre-calculating the thermal resistance according to various glazing systems. In subsequent studies, it would be necessary to consider this alternative method so as to make it easier to use the calculation method for the U-value of a double window. If the U-value of a double window is calculated according to ISO 10077-1, the result approximates the experimental value. This also overcomes the error associated with existing methods, which assume that the air cavity between internal and external windows is part of the glazing system. Therefore, the procedure detailed in ISO 10077 should be considered for the appropriate calculation of the thermal transmittance of double windows.

**Author Contributions:** Investigation, M.B. and Y.L.; Methodology, Y.L.; Project administration, J.K.; Validation, G.C.; Writing—original draft, M.B.; Writing—review & editing, S.K. All authors have read and agreed to the published version of the manuscript.

**Funding:** This research was supported by a grant (20RERP-C146906-03) from the Residential Environment Research Program funded by the Ministry of Land, Infrastructure, and Transport of the Korean government.

**Conflicts of Interest:** The authors declare no conflict of interest.

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
