# Peer review of "Analysis of the Calculation Method for the Thermal Transmittance of Double Windows Considering the Thermal Properties of the Air Cavity"

_sustainability, doi:10.3390/su122410439_

Round 1

Reviewer 1 Report

Even if all previous studies of the same authors are referenced (ref. n. 6, 7 and 8) in this article and even if in the 3 references, consider different aspects of the same subject, I want to suggest to the authors to revise the paper by excluding any type of overlapping. For example, figure 4 of the reference n.7 is quite identical to the figure 1 of this paper, except for the location of the thermocouples. 

In the present paper, the introduction section and related references literature must be improved.

Some other issues:

For the acronyms, you have to use capital letters (Ex: "Computational Fluid Dynamics" in the abstract)

In line 57, revise the repetition "acording to".

In lines 63-64, the sentence is a repetition of the previous sentence in lines 59-60.

In table 1, the glazing system D is composed by the same stratigraphy: 5CL + 12Air + 5LE, please revise, (maybe considering in D, "12Ar"?).

Reviewer 2 Report

The study presents the comparison of the thermal transmittance of double windows using laboratory measurments and methods provided by international standards, ISO 15099 and ISO 10077 (through CFD simulation). I have some concerns about the scientific soundness of the manuscript since the authors use referenced methods without a real improvement of them but following what the standards already require, ror example, CFD that how authors state is allowed by the international standard. From the title I would expected a real improvement of the current methods to calculate the termal performance of double windows.

I suggest the authors to deeply revised the amnuscript, including the relevant scientific literature on the topic, that at the moment lacks, especially in the introduction. The title must better reflect what the article deals with. The methods must be better described, especially the laboratory test. Results, discussion and conclusion must be considerably improved.

Reviewer 3 Report

Dear Authors,

In my opinion, your work is appropriately designed. The introduction, the part describing the research methods, the presentation of the U-value calculation, and the discussion of the obtained results, finally, the conclusions are consistent. Your paper will be understandable even for a reader who does not know the specified ISO and Korean standards. You presented clearly what the research problem is and how you solved it.

I have a few comments that, if you introduce them, will improve the quality of your work. Here they are:

  1. inline no. 57, you cite reference no. 7. Inline no. 69, the following references appear and are numbered 9 and 10 respectively, between lines 57 and 69, there is no reference no.8, which appears for the first time in line no. 162, Reference number 9 should be marked with 8, 10 should become 9, and 8 should become 10 to keep the chronological order.
  2. In a sentence (from lines 264 to 266) that reads: "In previous work, the importance of adopting the appropriate thermal properties of the air cavity between internal and external windows for calculation of the thermal performance of double windows was reported." the information in which the previous paper describes this issue should be inserted (6, 7 or 8? or other?).
  3. Reference number 3 - you should add a link to the website where this document is available.
  4. Reference number 4, the title of ISO 15099: 2003, should be: "Thermal performance of windows, doors and shading devices - Detailed calculations" and not what it is: "thermal performance of windows, doors and shading devices - detailed calculations."

Round 2

Reviewer 1 Report

The most part of my concerns have been solved.

Please, verify/explain what is the difference between glazing system B and D in Table 1 that have the same composition (5CL + 12Air + 5LE) but different Ug-value

Please revise also Table 4 that is not clear enough. I suggest to split it into two parts: the first table can be used to describe the different methods A, B, C. Then, in another table, can be reported the different values defined with the three methods and the comparison with the laboratory measurements.

Reviewer 2 Report

I would thank the authors for effort in improving the manuscript. I think that the experiment is carried out in a good manner by comparing calculation methodologies and experimental results. But in this kind of paper I would expect an improvement of the methodologies. Since ISO 15099 is the recognized standard I would expected a methodology, a framework, or something else to improve it. On the other hand, the authors report this statement in the conclusion. The results should be better analysed, for example what is the reason of the variability of the results from a window to another? In table 4 I suggest to express the percentage using the experimental results as reference, so the percentage is positive when the U-value of the calculation method is higher than the measured one, and negative when it is lower. 

English language needs improvement.

Round 3

Reviewer 2 Report

The authors respond to my comments, even if the concern about the improvement of ISO 15099 still exist.